# Candidate Waveforms for ARoF in Beyond 5G

**Javier Pérez Santacruz** [1],*[ID], **Simon Rommel** [1][ID], **Ulf Johannsen** [2][ID], **Antonio Jurado-Navas** [3][ID] **and Idelfonso Tafur Monroy** [1][ID]

1  Institute for Photonic Integration, Eindhoven University of Technology, 5600MB Eindhoven, The Netherlands; s.rommel@tue.nl (S.R.); i.tafur.monroy@tue.nl (I.T.M.)
2  Centre for Wireless Technology, Eindhoven University of Technology, 5600MB Eindhoven, The Netherlands; u.johannsen@tue.nl
3  Department of Communications Engineering, University of Málaga, 29071 Málaga, Spain; navas@ic.uma.es
*  Correspondence: j.perez.santacruz@tue.nl; Tel.: +34-681-230-250

**Abstract:** 5G mobile networks aim to support a large variety of services with different and demanding requirements. To achieve this, analog radio over fiber (ARoF) fronthaul along with millimeter-wave (mmWave) cells is a strong candidate to be part of the 5G architecture. Very high throughput can be achieved by using mmWave signals due to the large available bandwidths, which combines well with the advantages of employing ARoF technology. Nevertheless, combined mmWave and ARoF systems face a particular challenge as the impacts of both channels—such as high free-space path loss, phase noise, chromatic dispersion, and other degrading effects—affect the signal without the possibility for intermediate restoration. The selection of the signal waveforms plays an important role in reducing these defects. In addition, waveforms are one of the keys in the physical layer available towards satisfying the requirements for 5G and beyond. In this manuscript, several key requirements are presented to determine the merit of candidate waveform formats to fulfill the 5G requirements in the mmWave ARoF architecture. An overview of the different suitable waveforms for this architecture is provided, discussing their advantages and disadvantages. Moreover, a comprehensive comparison in terms of different requirements is also presented in this paper.

**Keywords:** 5G; ARoF; mmWave; DSP; waveform; modulation; OFDM

## 1. Introduction

The increasing number of mobile devices demanding internet applications has motivated the exploration of diverse possibilities and methods for achieving a higher capacity of exchanging information with enhanced coverage potential [1]. Based on ITU-T FG-IMT-2020 [2], fifth-generation (5G) networks should provide 1000 times more wireless capacity than currently available, supporting internet connectivity with exceptionally low latency (<1 ms) to over 7 trillion wireless devices among 7 billion people. Considering the capacity associated with the anticipated small cells, it is expected that data rate requirements range between 100 Mbit/s and 1000 Mbit/s and beyond, with peaks up to 10 Gbit/s.

Accordingly, 5G millimeter-wave (mmWave) wireless channel bandwidths will be more than ten times greater than current 4G Long-Term Evolution (LTE) cellular channels [3,4] to deliver an unprecedented level of service to the end user. Since wavelength shrinks by an order of magnitude at mmWave when compared to today's 4G microwave frequencies, they will be affected by a severe free space path loss (FSPL) and a considerable attenuation that is caused by diffraction and material penetration, thus elevating the importance of line-of-sight (LOS) propagation, reflection, and scattering. Therefore, cell coverage areas in 5G, are approximately in the range of 10 to 200 m [5]. This fact implies an increase in the number of cells and nodes in current mobile networks.

Fiber optic networks with their immense capacities are set to be the most important connection type for front- and backhaul for such wireless networks, due to their high bit rates achieved and the long distances covered. While in current centralized radio access network (C-RAN) deployments fronthaul data is transported in digitized form, i.e., as in phase and quadrature (IQ) samples of the RF waveform, this is highly inefficient with regards to optical spectrum usage and the required data rates quickly become prohibitive as radio bandwidth and carrier frequencies grow. In this respect, analog radio over fiber (ARoF) technology, where the required RF waveform is transported in the optical network as an analog signal, paves the way as an efficient solution in terms of spectral efficiency compared to digital RoF (DRoF) [4,6]. Thus, ARoF efficiently leverages on the advantages of fiber optics, such as low attenuation and high bandwidths.

To support the integration of diversified data traffic types and integration of mobile front- and backhaul with other services in a shared network, emerging flexible, robust and high capacity passive optical networks (PONs) are considered for ARoF architectures [7]. In addition, wavelength division multiplexing (WDM) is already commonly used in aggregation and metro networks because it provides a graceful upgrade path to accessing the available optical spectrum, as well as having advantages in terms of scalability and network management. As the number of required channels growth, especially with dense 5G deployments, the number of available wavelengths must be increased to allow provisioning of sufficient capacity while minimizing waste of spectrum. In this respect, the separation among wavelengths must be reduced, resulting in ultra-dense WDM (UDWDM). In this regard, the EU-H2020 project-ITN 5G STEP FWD [8] proposes to transform the current PONs to UDWDM-PONs.

The combination with ARoF, i.e., ARoF over UDWDM-PONs, is a strong candidate to be part of the 5G front- and backhaul architecture. As an illustrative case, Figure 1 shows a feasible structure of mmWave cells over UDWDM-PONS for RoF systems. However, the reduction of wavelength spacing in UDWDM-PONs and the impairments inherent to working with mmWave signals may induce system performance degradations [9]. Furthermore, the use of ARoF fronthaul directly concatenates the optical and wireless channels, creating a hybrid channel of potentially larger complexity than the pure wireless channel observed with digitized fronthaul. A possible way to mitigate impairments and to address a more complex channel lies in implementing advanced waveform formats.

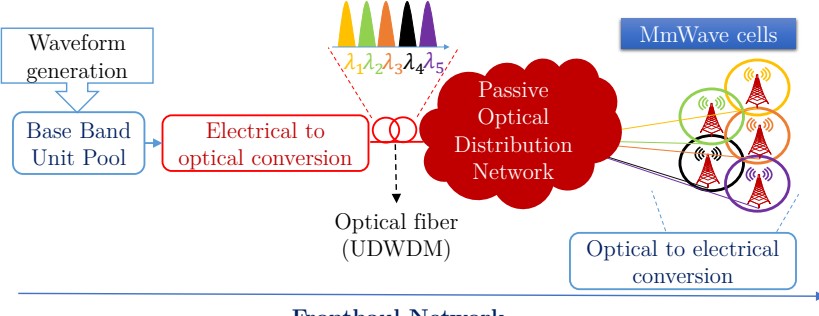

**Figure 1.** Radio over fiber (RoF) configuration for mmWave cells over ultra-dense wavelength division multiplexing-passive optical networks (UDWDM-PONs).

In terms of waveform, consider now LTE fourth-generation (4G) cellular networks, where orthogonal frequency-division multiplexing (OFDM) is the digital modulation technique adopted in the standards [10]. However, the spectral efficiency of OFDM is limited by the inclusion of a cyclic prefix (CP) and by its large side lobes, which require some null guard tones at the spectrum edges [11]. Furthermore, OFDM signals may suffer from large peak-to-average-power ratio (PAPR) values. Moreover, due to frequency deviations, the subcarriers will be no longer orthogonal, causing inter-carrier interference (ICI). Similar effects arise when the OFDM technique is affected by Doppler spread for the case of non-linear time-invariant (non-LTI) channels [12]. For those reasons, OFDM should be enhanced by using another alternative waveform format for next generation mobile

networks (5G and beyond). Thus, in this manuscript, we offer an analysis and comparison of different candidate waveform formats for future mobile networks.

This paper is organized as follows. Section 2 introduces the requirements for waveforms to be used in high-bandwith mmWave signals and ARoF transport. Section 3 shows and describes potential heirs of OFDM for beyond 5G in mmWave over UDWDM-PONs. Section 4 displays the state-of-the-art (SoA) on waveforms in ARoF experiments. Section 5 compares the presented candidate waveforms based on the requirements detailed in Section 2. Finally, Section 6 provides some concluding remarks.

## 2. Requirements to Waveforms for Beyond 5G

In this section, different key requirements associated with waveform formats for mmWave transport over ARoF systems are reviewed. Three main scenarios are distinguished: (i) general wireless communications, (ii) mmWave wireless communications, and (iii) ARoF systems. As shown in Figure 2, any requirement considered through this paper is connected with one or several 5G requirements, and thus they jointly allow a good comparison of candidate waveforms for 5G and beyond.

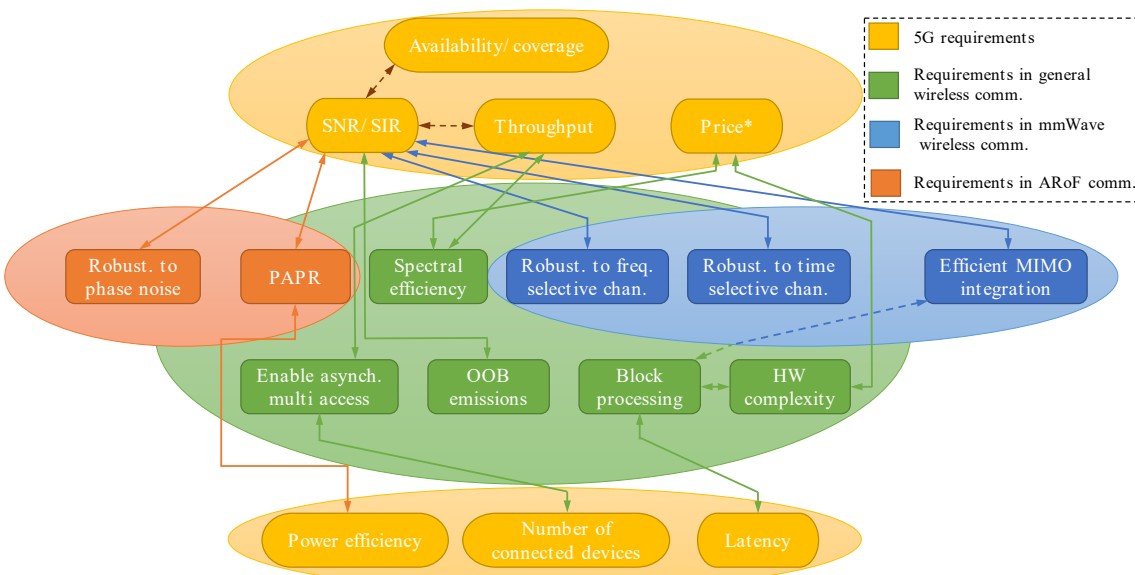

**Figure 2.** Relationships among requirements for candidate waveforms and 5G requirements. Three subsets of waveform requirements are defined: the ones for general wireless communications, the specific ones for mmWave wireless communications, and the ones for ARoF transport of high-bandwidth signals.

### 2.1. Requirements for General Wireless Communications

In terms of general requirements for wireless communications, a prospective waveform format should include the following features.

- Peak-to-average power ratio: PAPR indicates the relationship between the maximum peak and the average transmitted power of the signal. The worst impairment metrics associated to high PAPR in terms of communication are high power consumption and severe signal distortions. Note that energy efficiency is one of the most important requirements in 5G [13]. Signal distortions due to high power values are caused by nonlinearities in devices such as Mach–Zehnder modulators (MZMs) and power amplifiers, causing spectral regrowth and higher bit error rates (BER) [14]. Thus, a waveform format that produces high PAPR is not suitable for an energy efficient network.
- Spectral efficiency: Due to both licensing requirements and the spectrum scarcity resulting from the increasing transmission bandwidth requirement with demand for any time, anywhere, any situation communication this indicator is considered pivotal. The spectral efficiency is a very

important factor in a system because it is directly related to its bit rate achieved. According to the 5G requirements proposed in ITU-R M.2410-0 [13], the peak spectral efficient target is 30 bit/s/Hz and 15 bit/s/Hz for downlink and uplink respectively.

- Block processing delay: This requirement is seen relevant since it affects the final latency. A waveform format with high complexity suffers large block processing delays. Additionally, block processing delay is lower-bounded by symbol duration in many cases. The final latency of any communication can not be less than the block processing time. The latter can be reduced by employing techniques such as pipelining, efficient algorithms or by reducing symbol temporal period. One of the most challenging objectives in 5G is to reach communications with a maximum delay of 1 ms for the user plane [13].

- Robustness to frequency-selective channels: Multipath propagation is a phenomenon present in any wireless communication. It is caused by multiple reflections and refraction processes suffered by the transmitted signal, resulting in a received signal that is dispersed in time. Each path features its own delay and, accordingly, the temporal dispersion can induce to inter-symbol interference (ISI). Delay spread is a measure of the multipath profile of a mobile communications channel. As frequency fading can severely impact transmission, waveforms must be designed to be robust to this impairment.

- Robustness to time-selective channels: Most multipath channels are of time-varying nature. That nature arises as, for example, either transmitter the receiver are moving, and thus the location of reflectors in the transmission path, which gives rise to multipath, will change over time. Thus, if we repeatedly transmit pulses from a moving transmitter, we will observe changes in the amplitudes, delays, and the number of multipath components corresponding to each pulse. Regarding the 5G requirements proposed in ITU-R M.2410 [13], the 5G network should support a spectral efficiency of 0.45, 0.8, 1.12, and 1.5 bit/s/Hz for a mobility speed of 500, 120, 30, and 10 Km/h, respectively, and thus robustness to time-selective channels is key for candidate waveforms.

- Out of band (OOB) emissions: Linked to the spectral efficiency, this parameter is very significant as the radio spectrum is generally shared by different users, providers, and technologies. In order to efficiently support multiplexing of services, both in-band and out-of-band emissions must be kept to a minimum, so that services being transmitted on adjacent frequency channels do not interfere with one another. According to release 15 of 3GPP [15], the bandwidth is up to 400 MHz for carrier frequencies above 24 GHz. A portion of such a bandwidth (around 20%) is used as a guard band. Therefore, the OOB should be high enough to achieve a reduced interference between the adjacent channels and, thus, to obtain an adequate frequency multiplexing of services. For example, the OOB emission shall not exceed $-5$ dBm for bandwidths of 50, 100, 200 and 400 MHz in the OOB region of 0 to 5 MHz [15].

- Enabling asynchronous multiple access: Asynchronous multiple access is relevant as it allows to efficiently utilize resources. In frequency division duplex (FDD) and time division duplex (TDD) systems, asymmetric and dynamic allocation of both time and frequency resources is feasible for increasing bandwidths in order to accommodate the asymmetric traffic with higher efficiency [16]. Namely, waveform formats enabling asynchronous multiple access are connected with more efficient channel usage and corresponding higher total throughput.

- Filter granularity: This factor indicates the level in which the waveform is using the filtering stage. The filter granularity is directly related to latency and OOB emissions. As a direct consequence, long filters cause a high block processing delays and thus negatively impact achievable latency. On the contrary, other waveform formats implementing shorter filter lengths do not induce high latency because they filter by sub-band (wide filter bandwidth) [17]. Therefore, a trade-off between low OOB emissions and low latency is required. Thus, a very narrow filter granularity (subcarrier) implies very low OOB emissions. However, the filter length will be very long and, in consequence, the latency will increase.

- Hardware (HW) complexity: The importance of low hardware complexity is associated with both the final expense and the complexity of the system. As already mentioned in the introductory section, the number of cells will increase in the next generation of mobile networks. Therefore, the complexity and cost of the hardware in each cell is a key factor when determining the feasibility of a modulation format.

### 2.2. Requirements in MmWave Wireless Communications

MmWave signals are seriously affected by FSPL due to their inherent high frequencies. This fact plays an important role in determining the mmWave range. Furthermore, this type of signals are highly sensitive to attenuation. It turns out that atmospheric attenuation, rain-induced fading, snow, fog, foliage attenuation, and material penetration considerably adds more limitations to the maximum range of mmWave link [18]. Accordingly, mmWave signals reach shorter distances than the signals used in LTE. Fortunately, the reflected multipath components suffer a considerable attenuation so that their number is reduced [18]. Therefore, as the multipath effect in mmWave scenarios is less intense (except in special scenarios characterized by sand and/or dust atmosphere), it is more difficult to establish a non-line-of-sight (NLOS) communication [18]. Considering the limitation in distance that mmWave signals presents, a recommendable requirement for waveform formats is described below.

- Efficient MIMO integration: Multiple-input multiple-output (MIMO) systems are a suitable technique to overcome the aforementioned significant low attenuation of mmWave wireless communications. Massive MIMO is an extended solution to form very directive lobes in a certain direction. However, this technique demands high signal processing requirements to manage its associated beamforming matrix [19]. Therefore, a modulation format with efficient MIMO integration is required to reduce the complexity of the beamforming system.

### 2.3. Requirements in ARoF

ARoF combines optical and RF transmission. An example of a simple ARoF scheme is shown in Figure 3, where $f_{RF}$ is the RF carrier frequency and $f_L$ the optical carrier frequency. The characteristic of the spectrum form for each step of the ARoF system can also be observed in Figure 3. The appropriate requirements for this type of scheme are listed below.

- Robustness to phase noise: ARoF is limited by phase noise when phase modulations are used. In the optical part, one of the most prominent impairments of the optical fiber is the chromatic dispersion. This dispersion produces phase rotation and ISI. Furthermore, in the mmWave tone generation, phase noise is introduced. The impact of this phase noise depends on the used technique to produce the mmWave tones in the optical domain [20]. Therefore, high robustness to phase noise is a relevant requirement for a waveform format in an ARoF system.
- Dynamic range (DR): ARoF is restricted by dynamic range too. The DR determines the minimum and maximum amplitude of the signal received to recover the information correctly. Then, the maximum DR value is directly related to the highest signal peaks (PAPR). In the optical part, the noise floor is increased by relative intensity noise (RIN) from the laser, amplifier spontaneous emission (ASE) from the amplifiers, and thermal and shot noises from the photodiode [21]. In its part, each RF device adds noise that can be quantified by the noise figure. All these additive noise contributions increase the noise floor. On the other hand, a distortion region is created and increased by the non-linearity of the optical fiber and the RF amplifier [14]. This region is also incremented by the intermodulation products and spurious of the RF amplifiers and MZMs, respectively [22,23]. Thus, the distortion region and the noise floor, which suffer from ARoF systems, limit the DR extremely. Therefore, the DR of ARoF systems determines the type of waveform format that will be used and is related indirectly with the PAPR of the waveform.

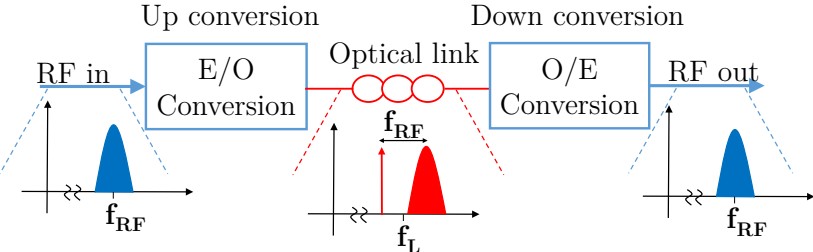

**Figure 3.** Simple analog radio over fiber (ARoF) scheme.

## 3. Candidate Modulation Formats for MmWave Over UDWDM-PONs

Waveform formats are mainly divided into two categories: multi-carrier and single carrier waveforms. Through this paper, we study different proposals belonging to any of both aforementioned groups. The transmitter structures for all the cases analyzed in this section are summarized in Table 1.

**Table 1.** Overview of the waveform schemes and their signal generation processes.

| | Constellation mapping | Up-sampling | Filter | FFT | IFFT | Sum | CP | Windowing | Filter |
|---|---|---|---|---|---|---|---|---|---|
| OFDM | ✓ | - | - | - | size $N$ | - | ✓ | +WOLA | bandpass (F-OFDM) |
| FBMC | OQAM | K factor per SR | per SR | - | size $KN$ | and overlap | - | ✓ | - |
| UFMC ($K$ SBs) | ✓ | - | - | - | $K$ IFFTs size $N$ (add zeros) | - | zero-guard (optional) | - | FL per SB and sum |
| GFDM | ✓ | per SR | per SR | - | - | ✓ | ✓ | +WOLA | - |
| MCAP | ✓ | per symb. | IQ FLs per SB | - | - | ✓ | - | - | - |
| SC-FDM | ✓ | - | - | size $N$ | size $> N$ (add zeros) | - | ✓ | +WOLA | - |
| SC amp. only | PM | - | PS ($\downarrow$ OOB) | - | - | - | - | - | - |
| SC amp. and phase | ✓ | - | PS ($\downarrow$ OOB) | - | - | - | - | - | - |

\* SR → subcarrier. FL → filter. SB → sub-band. IQ → in-phase and quadrature. PM → pulse mapping. PS → pulse shaping.

### 3.1. Orthogonal Frequency Division Multiplexing (OFDM)

OFDM is a popular multi-carrier waveform format developed for RF systems and applied in LTE downlink. It can significantly increase the data rate in bandwidth-constrained channels with high spectral efficiency and allowing efficient MIMO integration. Furthermore, the robustness of OFDM to either phase noise as to time-selective channels depends on the spacing of its subcarriers [24]. Therefore, by using the scalable numerology and the subcarrier spacing, OFDM can be robust to almost any channel condition imposed by the scenario and by other requirements.

On the other hand, the basic OFDM suffers high OOB emissions so multiple techniques have been proposed to reduce them. These techniques are classified into two main categories: (1) windowed-OFDM and (2) filtered-OFDM [12]. The weight overlap and add based OFDM (WOLA-OFDM) is the windowed technique implemented in LTE [25]. This technique greatly reduces the OOB emissions without including high complexity. The WOLA-OFDM OOB emission level decreases with the length of CP since this length determines the length of the window. An asymmetric window may be used instead of well-known symmetric windows for reduction of the cyclic prefix by 30 % [25] and, therefore, to reduce overhead. This technique suppresses OOB emission but makes the system more susceptible to channel induced ISI and ICI.

### 3.2. Filter Bank Multi-Carrier (FBMC)

FBMC is a multi-carrier waveform similar to OFDM that has been proposed in ITU-R M.2320 [16] as a promising waveform format for 5G. As the main feature, this waveform format does not include a CP. Therefore, its associated spectral efficiency is higher than that of OFDM. In addition, the half-Nyquist prototype filters mitigate ISI and the offset quadrature amplitude modulation (OQAM) removes ICI.

FBMC uses subcarrier filtering. Thanks to this feature, the filter length is long (high latency), the OOB emission are reduced, and the isolation between the subcarriers increases (high robustness to time-selective channels). However, a complex method is needed to estimate and compensate the channel. Therefore, MIMO integration with FBMC is more difficult. It is worth noting that FBMC can also employ scalable numerology.

### 3.3. Universal Filtered Multi-Carrier (UFMC)

UFMC is a type of sub-band filtering based on multi-carrier waveforms, combining the simplicity of OFDM with the advantages of FBMC. However, these advantages involve an increase in the complexity at the transmitter caused by the implementation of a filter and by applying a fast Fourier transform (FFT) for each sub-band, whereas at the receiver it requires doubling the size of the FFT. As a result, the total band composed of $N$ subcarriers is divided into $K$ sub-bands. Therefore, the UFMC transmitter performs the related $K$ inverse FFTs (IFFTs) of size $N$ separately, one per sub-band, by introducing zeros in the subcarriers not belonging to the sub-band. Each resulting signal is then filtered according to its frequency band and added to the other outputs [17].

On the other hand, the UFMC receiver recovers the signal through a size $2N$ FFT by adding zeros on the edge. Due to the size of the FFT, the use of CPs is avoided to correctly recover the signal, and thus a very high spectral efficiency can be achieved. However, this fact causes an additive noise increase in the receiver, thus obtaining a worse performance compared to OFDM [17].

### 3.4. Generalized Frequency Division Multiplexing (GFDM)

Like FBMC, GFDM is a multi-carrier waveform based on subcarrier filtering, where every subcarrier is shaped by a circular filter. The total number of mapped QAM symbols is arranged into $K$ subcarriers and $M$ subsymbols. Therefore, the total number of data symbols is $N = MK$ [26]. Next, every subcarrier is upsampled, filtered, and shifted to its carrier frequency. Then, these subcarrier signals are added, with a CP included at the end of each resulting block of subsymbols to avoid ISI [27]. Unlike OFDM, CPs are added per block (set of subsymbols) and not per symbol. Therefore, GFDM spectral efficiency is higher compared to OFDM [27], although with a high latency due to processing large blocks at a time.

As GFDM is not orthogonal, additional techniques must be implemented to properly recover the signal. There exist two main techniques: (1) interference cancellation scheme [28], and (2) OQAM [29], the latter with less complexity. Such techniques make GFDM receivers more complex. Moreover, GFDM is weaker than OFDM in terms of carrier frequency offset (CFO) [26].

### 3.5. Multi-Band Carrierless Amplitude and Phase Modulation (Multi-CAP)

Carrierless amplitude and phase modulation (CAP) is a multilevel and multidimensional modulation seen as a particular implementation of single carrier QAM using filters with orthogonal response. This absence of a carrier leads to less expensive and simpler transceivers compared to single carrier QAM, although increasing features in terms of spectral efficiency and performance [30,31].

As a representative feature, CAP is characterized by a low PAPR and simple implementation. However, CAP is proven to be very sensitive to frequency-selective channels and, to overcome this impairment, it requires a very complex equalizer and, consequently, suffers from inefficient MIMO integration. In this respect, a variant of CAP dividing the signal into different sub-bands, multi-CAP,

is proposed [30], where signal power and modulation order can be adapted to the concrete channel condition associated to each sub-band.

### 3.6. Single Carrier Frequency Division Multiplexing (SC-FDM)

The Single Carrier Frequency Division Multiplexing (SC-FDM) technique combines the advantages of OFDM, frequency-domain spread multi-carrier code-division multiple access (CDMA), and the conventional single-carrier direct-sequence CDMA (DS-CDMA). Moreover, interleaved frequency division multiple access (IFDMA) scheme in SC-FDM systems does not exhibit PAPR problems, while localized frequency division multiple access (LFDMA) implementation slightly conflicts in terms of PAPR. For these cases, we can maximize as much as possible that aforementioned figure of merit to achieve the best possible performance in the system.

In fact, SC-FDM and not OFDM is the waveform preferred in the LTE uplink due to its low PAPR and, in this respect, the energy consumption of the mobile station is hugely reduced. SC-FDM is not a pure single carrier waveform. Its features are halfway in between the features of pure single carrier waveforms and multi-carrier waveforms. Therefore, SC-FDM presents better performance than multi-carrier waveforms in terms of PAPR, but PAPR is higher than that of pure single carrier waveforms [24]. Like basic OFDM, SC-FDM has high OOB emissions. The WOLA technique is further applied in SC-FDM in order to reduce the level of secondary lobes affecting adjacent bands.

### 3.7. Single Carrier Amplitude Only

Referring to single carrier waveforms that solely modulate the signal by amplitude, this type of waveform is characterized by its simplicity and robustness to phase noise. As weak points, they shows very low spectral efficiency and high OOB emissions. As a representative implementation of single carrier amplitude only, widely employed in optical systems, we can mention on–off keying (OOK). Pulse with modulation (PWM) and pulse position modulation (PPM) are other types of single carrier amplitude modulation that encode each symbol with different pulse width and pulse position, respectively. OOK, PWM, and PPM provide very low spectral efficiency. To achieve a higher value, several techniques were proposed. One of these consists of adding several levels of amplitude to the resulting modulated symbols. In this way, pulse amplitude modulation (PAM), multilevel-PWM (M-PWM) and vector weight multilevel PPM (vw-MPPM) [32] arise.

Finally, an effort to reduce the huge OOB emissions inherent to pure single carrier waveforms is to generate Gaussian pulses instead of rectangular pulses. Furthermore, the average power is reduced with this mechanism and so, the PAPR magnitude.

### 3.8. Single Carrier Amplitude and Phase

This solution supposes higher order modulation than the waveforms explained above. Its main features are the same as for a single carrier waveform. The modulation formats employed are based on amplitude and phase. Therefore, high spectral efficiency can be achieved. However, these systems are, in general, weaker to phase noise.

One of the most popular amplitude and phase modulation formats is QAM. Based on it, amplitude and phase shift keying (APSK) was proposed and, as QAM, it is considered a combination of amplitude shift keying (ASK) and phase shift keying (PSK), but without being restricted to quadrature constellations. Namely, APSK is more flexible than QAM and, thus, there are many APSK constellation designs for different channels [12] with remarkable importance for those focused on achieving a channel capacity very close to the Shannon limit.

## 4. SoA of Waveforms Used in ARoF

Next, in this section, we review the use of waveform formats in ARoF research and experiment. The attention of ARoF research has been devoted to merging radio frequency and optical fiber technologies, aiming to increase the capacity and mobility of the access network. In the first attempts,

ARoF setups employed single carrier waveforms for their simplicity and robustness. Waveforms such as OOK and single carrier QAM were used in [33–38], respectively. However, they are not optimal for wireless communications due to their limited spectral efficiency and the scarcity of spectrum in the wireless channel.

Once OFDM was proposed for 4G, experiments involving ARoF also started employing it, and nowadays this waveform is still widely used since it is included in the standard for the current mobile network and for the first release of 5G [15]. ARoF setups with OFDM are found in [3,39–43] as illustrative examples. However, OFDM presents several issues as already discussed above and, for that reason, advanced multi-carrier waveforms have emerged recently such as FBMC, UFMC, GFDM and multi-CAP, which can be prominent candidates beyond 5G. Accordingly, these waveforms are now used in ARoF experiments: FBMC in [44,45]; GFDM in [46–49]; and multi-CAP in [50–52], to name but a few.

In Figure 4 we have compiled the recent usage trend of the different waveforms considered here in ARoF experiments of the last ten years. In this respect, ARoF set-ups have been organized in terms of wired/wireless experiments, real-time or offline signal processing, and featuring carriers below or above 6 GHz. Remarkably, as indicated in Figure 4, OFDM and single carrier waveforms are the most employed waveform formats for the considered period of the last ten years. Nevertheless, advanced multi-carrier waveforms are acquiring increasing relevance to mitigate ISI and multipath impairments.

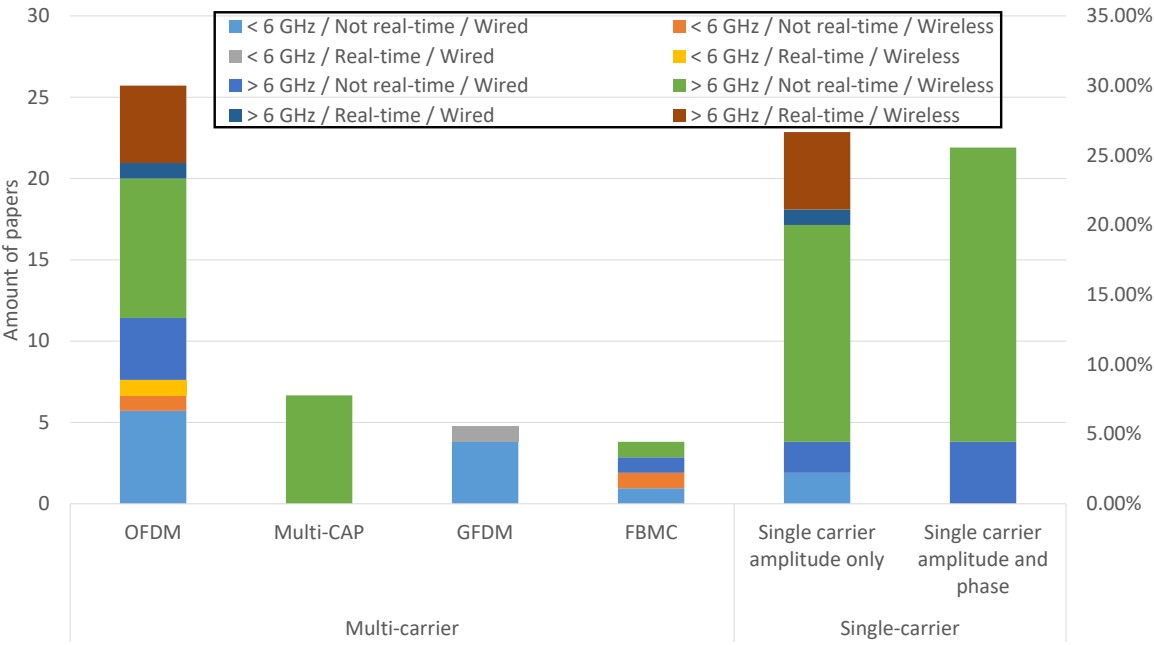

**Figure 4.** Overview about the state-of-the-art (SoA) on the waveform formats in ARoF experiments corresponding to the last ten years.

The aim of some of these experiments is to emulate the behavior of a mmWave cell over an ARoF system for 5G. To correctly emulate such a case, it is necessary to use carrier frequencies above 6 GHz, including the wireless link and implement real-time signal processing (brown label). However, from Figure 4 we observe that only OFDM and single carrier modulation formats have been used under these conditions. Therefore, for the rest of the waveform formats, research under such complete conditions, especially including real-time signal processing and the inclusion of the mmWave wireless transmission, is required to fill this gap.

On a further note, the future 5G will work with bands below and above 6 GHz for wireless transmission. Therefore, performance evaluations for the identified candidate waveforms in these

bands are required and a comprehensive comparison thereof is of great value in identifying the best candidates for beyond 5G. In this respect, these performance evaluations should include real-time signal processing to judge on their complexity and feasibility in as system context, as well as include actual wireless transmission to evaluate them under realistic channel conditions.

## 5. Comparison of Candidate Waveforms

Table 2 summarizes the different waveform formats analyzed throughout this paper in terms of the requirements discussed in Section 2. Thus, for each of those waveforms, different requirements are compared and analyzed in further detail.

- Filter granularity: Both GFDM and FBMC need longer filter lengths as they perform it by subcarrier (narrow filter bandwidth). On the other hand, F-OFDM, UFMC, and multi-CAP use shorter filter length because its granularities are per sub-band (wide filter bandwidth) [17]. Filter granularity can only be associated with multi-carrier waveforms (full band, sub-band, and subcarrier). Therefore, GFDM and FBMC present higher latency that the rest of multi-carrier waveforms.

- PAPR: From Table 2, we can observe that pure multi-carrier waveforms (OFDM, FBMC, UFMC, and GFDM) are associated with high values of PAPR. On the contrary, single-carrier waveforms present very low PAPR. It is worth noting that multi-CAP and SC-FDM both provide a low PAPR because they are not pure multi-carrier waveforms. On the other hand, any multi-carrier waveform can reduce its PAPR through different techniques. Nevertheless, they all increase the complexity in the system and, furthermore, include at least one of the following impairments [53]; power increase, bandwidth expansion, or BER degradation. For this reason, these techniques are not frequently considered as desirable.

- Spectral efficiency: This indicator is very important in order to achieve the bit rate requirements for 5G and beyond. High spectral efficiency is reached with a multi-carrier waveform and it increases by augmenting the modulation order. Single-carrier waveforms on the other hand provide lower spectral efficiency due to their limitations in the spectral domain. Comparing the spectral efficiency among the multi-carrier waveforms, FBMC, GFDM, and UFMC are the best because of their CP structure. In particular, FBMC and UFMC do not use CP while GFDM requires low CP overhead.

- Block processing delay: GFDM and FBMC present large block processing delays, as, among other reasons, its filter lengths are long, as mentioned previously. On the other hand, the block processing for the single-carrier waveforms is low because of their simplicity. Moreover, it is important to highlight that the final delay is intrinsically related to the symbol duration and thus, the latency is proportional to the subcarrier spacing in the case of FFT-based modulation formats.

- Robustness to phase noise: Single-carrier amplitude only is the best option as it does not use phase modulation. Robustness to phase noise is a pivotal point for low-cost base stations as very sophisticated and expensive devices are necessary to reduce the phase noise in hardware [24]. Pure single-carrier waveforms are inherently robust to phase noise, and they are better than the multi-carrier waveforms in this aspect [24]. However, we have to consider that this type of robustness is proportional to spacing among the subcarriers in the multi-carrier waveforms as mentioned in Section 3.

- Robustness to frequency-selective channels: Multi-carrier waveforms are better than single-carrier waveforms for this factor because frequency-selective fading will affect only a few subcarriers and not the entire band. That is, with adaptive bit loading, the impact of frequency-selectivity can be normalized.

- Robustness to time-selective channels: Single-carrier waveforms present better behavior than multi-carrier waveforms since the ICI inherently affects the multi-carrier waveforms [24]. Furthermore, as explained in Section 3, this robustness is proportional to the spacing among the subcarriers in the multi-carrier waveforms. GFDM, in particular, is the worst option due

to it needs long symbol duration, and therefore the changes of the channel strongly affect the GFDM symbol. On the other hand, and for the case of multi-carrier waveforms, FBMC is the best solution since the much better frequency-domain localization for the transmit filter than any other multi-carrier waveform. Therefore, the ICI can be efficiently removed for each subcarrier [17].

- OOB emissions: Because of their configuration, the OOB emissions of multi-carrier waveforms are much lower than those of single-carrier waveforms. At this point, OOB emissions can be reduced via filtering or pulse shaping. The characteristic of these techniques strongly influence the final OOB emissions. FBMC provides the lowest OOB emissions due to its filtering by subcarrier.

- Efficient MIMO integration: This factor has a strong relationship with the channel equalization implemented in the system. In addition, it is an indicative of the complexity of MIMO systems. All pure multi-carrier waveforms present highly efficient MIMO integration because they do not use complex channel estimation. Specifically, they use frequency-domain channel estimation through equally spaced pilots. FBMC and GFDM are exceptions in this case, as they require more complex channel equalization. Namely, FBMC needs to eliminate the imaginary interference in each scattered pilot [54], while GFDM requires a channel estimation in each subsymbol. On the other hand, single-carrier waveforms need more complex channel estimation to compensate. A popular estimation technique for this type of waveform is the adaptive decision feedback equalizer (DFE). It is a complex estimation and that is why the single-carrier waveforms are less efficient in terms of MIMO integration.

- Enable asynchronous multiple access: Pure multi-carrier waveforms do not allow to implement asynchronous multiple access in the system. This is due to the use of slots distributed in frequency, not in time. Consequently, this was one of the main reasons why SC-FDM was selected to be the waveform in the uplink for LTE.

- HW complexity: To implement the waveform in a field-programmable gate array (FPGA), the complexity of the system will determine the needed number of slice registers, look-up tables (LUTs) and random access memory (RAM) blocks [27]. According to Table 1, the single-carrier waveforms present a lower HW complexity as they need a smaller number of operations in order to process the transmitted and received signal. On the other side, GFDM, UFMC, and FBMC are the most complex waveforms due to the additional procedures that they add.

**Table 2.** Comparison between different waveform formats for mmWave ARoF.

| Requirement\Waveform | OFDM | FBMC | UFMC | GFDM | MCAP | SC-FDM | SC amp. Only | SC amp. and Phase |
|---|---|---|---|---|---|---|---|---|
| Filter granularity | FB | SR | SB | SR | SB | - | - | - |
| PAPR | H | H | H | H | L | L | VL | VL |
| Spectral efficiency | H | VH | VH | VH | M/H | H | M/L | M/H |
| Block processing delay | M | H | M | H | M/L | M | L | L |
| Robust. to phase noise | M | M | M | M/H | M/H | M/H | H | M/H |
| Robust. to freq.-selec. chan. | H | H | H | H | M/H | M/H | M | M |
| Robust. to time-selec. chan. | M | M/H | M | M/L | M | M/H | H | H |
| OOB emissions | L | VL | VL | L | L | L | M/H | M |
| Efficient MIMO integration | H | M | H | M | L | M | L | L |
| Enable async. multi. access | No | No | No | No | Yes | Yes | Yes | Yes |
| HW complexity | M | H | H | H | M/L | M/H | L | L |

* FB → full-band.　SR → subcarrier.　SB → sub-band.　V → very.　H → high.　M → medium.　L → low.

Observing Table 2, we can conclude that it is difficult to decide which waveform can be more suitable as they offer different advantages and disadvantages. In fact, any of those waveforms are well-appropriated for a particular type of service or channel due to their capacity to adapt to a subgroup of specific requirements. Indeed, according to IMT-2020, there are three foreseen categories of traffic types in 5G and beyond. These have different characteristics and use cases [55]: enhanced mobile broadband (eMBB) mainly requires high bit rate, massive machine-type communications (mMTC) will

support a huge quantity of devices with low power requirements, and ultra-reliable and low latency communications (URLLC) are targeted at mission critical communications where both low latency and superior reliability must be guaranteed. Thus, the optimal solution would be to select a suitable waveform format for each scenario for both uplink and downlink. It is necessary to highlight that an additional requirement is crucial in the uplink, resulting from the fact that the user performs a multicast transmission to the base station. Thus, to enable asynchronous multiple access is a primordial requisite for the uplink and hence, waveforms allowing asynchronous multiple access are very recommendable in the uplink.

For eMBB, the main requirement is the bit rate. Therefore, the spectral efficiency of the selected waveform should be very high and it should be robust to the ARoF system to achieve high modulation orders. In this sense, FBMC and UFMC are the two selected candidates since their characteristics are better adapted to those requirements. Nevertheless, FBMC implies a complex MIMO scheme due to its method to estimate and compensate the channel. In its part, UFMC allows an easier MIMO system to accomplish high antenna gain and, in consequence, high signal-to-noise ratio (SNR). Therefore, UFMC is one of the best candidates to be the heir of OFDM in the eMBB downlink. Concerning the uplink, the waveforms that allow asynchronous multiple access and high spectral efficiency are multi-CAP and SC-FDM. Indeed, multi-CAP can further achieve a higher bit rate because it can adapt the modulation order depending on the SNR in each sub-band [31]. Thus, multi-CAP could be the heir of the SC-FDM for the uplink in eMBB.

Following the mMTC requirements, the used waveform should support a huge number of devices. These devices transmit reduced amount of information as indicators of the system. Therefore, the bit rate is not a critical condition in this case. In mMTC, simplicity is a very relevant requirement because power consumption is of utmost importance and most devices have limited HW and SW. Sensors are a typical example for this type of scenario. Regarding the downlink in mMTC, OFDM is a proper candidate due to its simplicity, robustness to the ARoF system and capability to have smooth MIMO integration. For the uplink, multi-CAP and SC-FDM present better resilience to the ARoF impairments than the single carrier waveform options. Nevertheless, multi-CAP demands complex channel equalizer structure and it increases complexity for an ARoF scheme. Furthermore, its less efficient MIMO integration constitutes a problem in a system where there is a huge quantity of users employing the resources. Hence, SC-FDM still being the best option for the mMTC uplink, like in LTE.

Focusing on the main requirement of URLLC, pure single-carrier waveforms are the best solution to reach extremely low latency because of their low block processing delays. On the other hand, the reliability can be achieved by adapting the modulation order according to the channel conditions. We can reduce the modulation order because the bit rate is not critical in this scenario. SC amplitude and phase waveforms allow more flexibility to change the modulation order compared to SC amplitude only waveforms. Therefore, SC amplitude and phase waveforms are good candidate waveforms in URLLC both for downlink and uplink.

An alternative solution could consist in adaptive waveform formats according to the type of service. Namely, to select the waveform that best adapts to the requirements of a particular service. This can be achieved through intelligent software-defined radio (SDR) [49] and software-defined networking (SDN) [56].

## 6. Conclusions

In this paper, we have briefly described the prospective waveform candidates for 5G and beyond. In this regard, we have presented the main key requirements determining the performance of the waveforms in an ARoF system. Next, we have presented the current SoA of waveforms used in ARoF experiments, indicating the trend in the last ten years. In this SoA, and to the best of our knowledge, a deep comparison involving all waveform candidates for 5G with regard to ARoF was missing. In this work, we present such an SoA in terms of the requirements for ARoF based systems. Even so,

it remains difficult to select a potential best candidate for all situations. For that reason, IMT-2020 defined three types of scenarios [55].

Furthermore, in this paper, we have concluded the best waveform candidates for eMBB, mMTC, and URLLC for both downlink and uplink, respectively. The waveforms have to enable asynchronous multiple access in the uplink. Concerning eMBB, we conclude UFMC is one of the best solutions to achieve the highest bit rate in the downlink. On the other side, multi-CAP could be a very considerable successor of SC-FDM in the eMBB uplink. Regarding mMTC, we have remarked that simplicity is a considerable factor due to the SW and HW limitations of the devices in this type of scenario. Therefore, we have determined that OFDM and SC-FDM are the best waveforms because of their low complexity in downlink and uplink, respectively. Finally, for URLLC, pure SC waveforms could be the best candidates both for uplink and downlink to minimize latency. In particular, SC amplitude and phase modulation is one of the best options because it provides very low latency and flexibility to adapt the modulation order depending on the channel conditions.

At this point, it is important to highlight that the waveforms of the comparison presented are under different conditions. Therefore, an experimental comparison between all these modulation formats in an ARoF set-up with wireless transmission would be required to perform a fully comprehensive performance analysis. This experiment should be divided into three steps referring to each scenario. Each waveform should be adapted to achieve the best result in terms of the requirements of each scenario. In other words, the parameters of each waveform are modified to achieve the maximum bit rate for eMBB, the massive device support for mMTC and the lowest latency for URLLC. Therefore, the best candidate could be identified under realistic conditions and with an equal playing field to select the most promising waveforms for the mobile communication standards beyond 5G.

**Author Contributions:** J.P.S., S.R., and U.J. defined the scope of the review manuscript. J.P.S. compiled the necessary information to elaborate this review manuscript. J.P.S., S.R. and A.J.-N. wrote the article. J.P.S., S.R., A.J.-N., U.J. and I.T.M. reviewed and edited the manuscript. All authors have read and agreed to the published version of the manuscript.

**Funding:** This research was funded by the European Commission through H2020 ITN 5G STEP FWD (grant agreement 722429) and H2020 blueSPACE (grant agreement 762055) projects.

**Conflicts of Interest:** The authors declare no conflict of interest.

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
