# Peer review of "Candidate Waveforms for ARoF in Beyond 5G"

_applsci, doi:10.3390/app10113891_

Round 1
Reviewer 1 Report
This paper is a very comprehensive review about modulation formats for the next 5G and beyond communications.
I would like to thank the authors for their contribution, since it was a pleasure to read it.
I just found some typos in the document:
- Page 2, line 33: "type for front- and backhaul to for such..."
- Page 2, line 66: "Moreover, frequency deviation the subcarriers will be no longer orthogonal"... Maybe: "Moreover, due to frequency deviations, the subcarriers will be no longer orthogonal".
- Page 6, Figure 3: If fL is the optical carrier frequency, it should be the frequency value where the delta of the signal in the optical link is. In the center plot of this Figure, the label fL is at some frequency between the delta and the frequency components of the signal.
Reviewer 2 Report
Javier Pérez Santacruz et al. present a review paper entitled “Candidate Waveforms for ARoF in Beyond 5G” in which they have thoroughly analysed and compared different candidate waveforms for next-generation networks (5G and beyond). Specifically, they provide an overview of different suitable waveforms, discussing their advantages and disadvantages in terms of different requirements, to be employed with analog radio over fiber and milllimeter-wave cells.
In general, the paper is very well written and it provides a reach and interesting discussion for the scientific community. In my opinion, the paper can be published in Applied Sciences in the current form.
